# Disrupted Mitochondrial and Metabolic Plasticity Underlie Comorbidity between Age-Related and Degenerative Disorders as Parkinson Disease and Type 2 Diabetes Mellitus

**DOI:** 10.3390/antiox9111063

**Published:** 2020-10-30

**Authors:** Diana Luz Juárez-Flores, Mario Ezquerra, ïngrid Gonzàlez-Casacuberta, Aida Ormazabal, Constanza Morén, Eduardo Tolosa, Raquel Fucho, Mariona Guitart-Mampel, Mercedes Casado, Francesc Valldeoriola, Joan de la Torre-Lara, Esteban Muñoz, Ester Tobías, Yaroslau Compta, Francesc Josep García-García, Carmen García-Ruiz, Jose Carlos Fernandez-Checa, Maria José Martí, Josep Maria Grau, Francesc Cardellach, Rafael Artuch, Rubén Fernández-Santiago, Glòria Garrabou

**Affiliations:** 1Laboratory of Muscle Research and Mitochondrial Function, Department of Internal Medicine-Hospital Clínic of Barcelona (HCB), Institut d’Investigacions Biomèdiques August Pi i Sunyer (IDIBAPS), Faculty of Medicine and Health Science, University of Barcelona (UB), 08036 Barcelona, Spain; ingrid.gonzalez.casacuberta@gmail.com (ï.G.-C.); cmoren1@clinic.cat (C.M.); mg1616@camden.rutgers.edu (M.G.-M.); joan.delatorre7@gmail.com (J.d.l.T.-L.); etobiasb@ub.edu (E.T.); fjgarcia@ub.edu (F.J.G.-G.); jmgrau@clinic.cat (J.M.G.); fcardell@clinic.cat (F.C.); 2Centro de Investigación Biomédica en Red (CIBER) de Enfermedades Raras (CIBERER), 28029 Madrid, Spain; aormazabal@sjdhospitalbarcelona.org (A.O.); mcasado@hsjdbcn.org (M.C.); rartuch@sjdhospitalbarcelona.org (R.A.); 3Laboratory of Parkinson Disease and Other Neurodegenerative Movement Disorders, IDIBAPS, Institut de Neurociències, UB, 08036 Barcelona, Spain; ezquerra@clinic.cat (M.E.); etolosa@clinic.cat (E.T.); fvallde@clinic.cat (F.V.); jemunoz@clinic.cat (E.M.); ycompta@clinic.cat (Y.C.); mjmarti@clinic.cat (M.J.M.); rfernan1@clinic.cat (R.F.-S.); 4Centre for Networked Biomedical Research on Neurodegenerative Diseases (CIBERNED CB06/05/0018), 28029 Madrid, Spain; 5Department of Clinical Biochemistry, Institut de Recerca de Sant Joan de Deu, Esplugues de Llobregat, 08036 Barcelona, Spain; 6Parkinson Disease and Movement Disorders Unit, Neurology Service, HCB, IDIBAPS, UB, 08036 Barcelona, Spain; 7Department of Cell Death and Proliferation, Instituto de Investigaciones Biomédicas de Barcelona (IIBB), Consejo Superior Investigaciones Científicas (CSIC), 08001 Barcelona, Spain; raquel.fucho@iibb.csic.es (R.F.); carmen.garcia@iibb.csic.es (C.G.-R.); josecarlos.fernandezcheca@iibb.csic.es (J.C.F.-C.); 8Liver Unit-HCB, IDIBAPS, 08036 Barcelona, Spain; 9USC Research Centre for ALPD, Keck School of Medicine, Los Angeles, CA 90033, USA

**Keywords:** T2DM (type 2 diabetes mellitus), iPD (idiopathic Parkinson’s disease), mitochondria, metabolome

## Abstract

Idiopathic Parkinson’s disease (iPD) and type 2 diabetes mellitus (T2DM) are chronic, multisystemic, and degenerative diseases associated with aging, with eventual epidemiological co-morbidity and overlap in molecular basis. This study aims to explore if metabolic and mitochondrial alterations underlie the previously reported epidemiologic and clinical co-morbidity from a molecular level. To evaluate the adaptation of iPD to a simulated pre-diabetogenic state, we exposed primary cultured fibroblasts from iPD patients and controls to standard (5 mM) and high (25 mM) glucose concentrations to further characterize metabolic and mitochondrial resilience. iPD fibroblasts showed increased organic and amino acid levels related to mitochondrial metabolism with respect to controls, and these differences were enhanced in high glucose conditions (citric, suberic, and sebacic acids levels increased, as well as alanine, glutamate, aspartate, arginine, and ornithine amino acids; *p*-values between 0.001 and 0.05). The accumulation of metabolites in iPD fibroblasts was associated with (and probably due to) the concomitant mitochondrial dysfunction observed at enzymatic, oxidative, respiratory, and morphologic level. Metabolic and mitochondrial plasticity of controls was not observed in iPD fibroblasts, which were unable to adapt to different glucose conditions. Impaired metabolism and mitochondrial activity in iPD may limit energy supply for cell survival. Moreover, reduced capacity to adapt to disrupted glucose balance characteristic of T2DM may underlay the co-morbidity between both diseases. Conclusions: Fibroblasts from iPD patients showed mitochondrial impairment, resulting in the accumulation of organic and amino acids related to mitochondrial metabolism, especially when exposed to high glucose. Mitochondrial and metabolic defects down warding cell plasticity to adapt to changing glucose bioavailability may explain the comorbidity between iPD and T2DM.

## 1. Introduction

In Western countries, increased life expectancy has led to a rise in chronic and age-related diseases, resulting in a decrease in quality of life of the elderly population and representing an important sociosanitary burden [1,2,3]. Parkinson’s disease (PD) and diabetes mellitus (DM) are among the most prevalent 21st century epidemics. The world prevalence of PD is 1% of the population over 65 years, with an expected 50% increase for 2040 [4]. The world prevalence of diabetes was estimated to be 6.4% in 2010 and is expected to rise to 10.4% in 2040, of which 90% is represented by type 2 DM (T2DM) [5]. Clinical manifestations in idiopathic PD (iPD) and T2DM are secondary to the decrement of biological products as a result of cell death in the target tissue of the disease. Both diseases affect multiple organs and molecular alterations appear years before clinical diagnosis is made [6,7,8]. Thus, for either iPD or T2DM, preventive measures and disease modifying therapies are a main target in research [7].

PD is characterized by the decrease of dopamine release, resulting from the loss of dopaminergic neurons in the *substantia nigra*, associated with movement disorders. The main risk factor for PD is ageing and, despite the small proportion of PD cases of genetic origin (3–10%), in the majority of cases, PD is a result of several environmental factors, such as exposure to pesticides, brain injury, use of beta blockers, and well-water drinking [9].

T2DM is characterized by chronically elevated blood glucose concentration, which arises from a combination of insufficient insulin secretion and a reduced sensitivity of target cells and tissues to insulin. Obesity and sedentarism are the main risks factor for T2DM, and approximately 50% of diabetics show additional complications by the time they are diagnosed [10].

The comorbidity of iPD and T2DM has been described in numerous epidemiological studies [11,12,13]. Despite initial controversy [14,15], the established risk for presenting iPD in T2DM patients is 38% [16]. It is also known that T2DM may predispose to a PD-like pathology and induce a more aggressive phenotype when coexisting with PD [17]. In fact, T2DM is a well-known established risk factor for developing iPD [12].

Recent evidence points toward shared deregulated molecular pathways between PD and DM such as protein misfolding and depot, endoplasmic reticulum stress, oxidative damage, or inflammation [13,18,19].

Among these, growing evidence supports metabolic and mitochondrial dysfunction to play a critical role in the development of both diseases [11,12,13,15,18,20]. Specifically, mitochondrial respiratory chain (MRC) dysfunction, oxidative stress, altered mitochondrial dynamics, and morphology have been reported in iPD [21]. Similarly, insulin resistance is associated with increased oxidative stress, mitochondrial DNA mutations, and mitochondrial dysfunction [18]. In fact, mitochondria are the focus of preclinical therapies as pivotal players in the development for T2DM and iPD [22].

While metabolic and mitochondrial dysfunction clearly play a role in both iPD and T2DM, scarce studies have explored if such metabolic and mitochondrial deregulation may underlie the epidemiologic co-morbidity that exists between both diseases [12].

There is a crucial need to fully understand the molecular basis of the epidemiologic comorbidity between iPD and T2DM and if it may be associated with a further worsening of the bioenergetic deficits characteristic of both diseases.

Taking advantage of the validation of fibroblasts as a reliable cell model for the study of PD [23,24,25,26], we have designed an in vitro model to test weather hyperglycaemia characteristic of T2DM may worsen metabolic and mitochondrial phenotype contributing to reported comorbidity between PD and T2DM.

## 2. Materials and Methods

### 2.1. Study Design and Population

A single-site, cross-sectional, observational study was conducted. Fourteen age and gender paired subjects were included: seven iPD patients and seven healthy unrelated controls (C). No significant differences in age and gender were observed between cases and controls (Table 1). All iPD patients met the U.K. Brain Bank Criteria for the diagnosis of PD [27]. Subjects with other comorbidities (including T2DM) were excluded from the study [28]. Both patients and controls signed the informed consent to participate in the study, previously approved by the Clinical and Research Ethical Committee of our institution, in accordance with the Declaration of Helsinki (code HCB/2015/0562).

### 2.2. Fibroblasts Culture

Fibroblasts were obtained from the alar surface of the nondominant arm of the subjects by a 6 mm skin punch biopsy and mutation screening was performed to discard genetic contribution to the disease, as previously described [29].

Cells were grown in 25 mM glucose Dulbecco’s Modified Eagle’s medium (DMEM from Gibco, Life Technologies™, Burlington, ON, Canada) supplemented with 10% heat-inactivated fetal bovine serum and 1% penicillin-streptomycin at 37 °C, in a humidified 5% CO_2_ air incubator, until 80% optimal confluence was reached. After cell expansion, cells were exposed for 10 days to two different glucose conditions to assess metabolic and mitochondrial contribution to disease: a ‘pre-diabetogenic’ high glucose (HG) environment containing 25 mM of glucose (equivalent to 450 mg/dL) or to a normoglycemic environment containing 5 mM of glucose (equivalent to 90 mg/dl; low glucose, LG). Fresh media was provided every third day (three times in total in the 10-day lapse period), and cells were prevented from reaching more than 80% confluence. Fibroblasts were then harvested with 2.5% trypsin (Gibco, Life technologies™, Burlington, ON, Canada and centrifuged at 500× *g* for 8 min for further analysis.

Metabolic and mitochondrial phenotyping was performed in iPD and C fibroblasts between passage 5 and 10. The oxygen consumption rate (OCR), which requires assessment of live cells, was performed in parallel including iPD and C in both glucose concentrations. Fixation of cells for immunofluorescent quantification of mitochondrial dynamics was also performed with fresh material at this time point. Cell pellets were kept at −80 °C until the rest of the experimental procedures were performed.

### 2.3. Targeted Metabolomic Characterization

Fibroblast preparation for metabolomic assessments required a minimum of 1 million cells, which were thawed, resuspended in 200 µL of phosphate buffered saline (PBS), and centrifuged (1500× *g*; 10 min) to collect the supernatant, where amino acids and organic acids were quantified.

Organic acids were extracted in fibroblasts with ethyl acetate and diethyl ether and derivatized with bis(trimethyl-silyl) trifluoro-acetamide (BSTFA), as previously reported [30]. The trimethylsilyl derivatives obtained were separated by gas chromatography (Agilent 7890A, Wilminton, DE, USA) and detected in a mass spectrometer (Agilent 5975C, Wilmington, DE, USA). The results were expressed as nanomoles of organic acid per milligram of protein (nmol/mg protein).

Amino acids were quantified in fibroblasts by ultra-performance liquid chromatography (UPLC) coupled to tandem mass spectrometry, as previously reported [31]. Briefly, amino acids were separated in a Waters ACQUITY UPLC H-class chromatograph and quantified with a Waters Xevo TQD triple-quadrupole mass spectrometer using positive electrospray ionization conditions in the MRM (multiple reaction monitoring) mode. The results were expressed as nanomoles of amino acid per milligram of protein (nmol/mg protein).

### 2.4. Mitochondrial Characterization

MRC and citrate synthase (CS) enzyme activities were measured at 37 °C by spectrophotometry, following standardized procedures [32], as reported elsewhere [8,33]. All enzymatic activities were run in parallel with internal quality controls. Complex II (CII), Complex IV (CIV), and glycerol-3-phosphate dehydrogenase (G3PDH) enzyme activities were then normalized by CS to express enzymatic activities per mitochondrial content, as CS is widely considered as a reliable marker of mitochondrial mass. Changes in absorbance were registered in a HITACHI U2900 spectrophotometer through the UV-Solution software v2.2 (Tokyo, Japan) and expressed as nanomoles of consumed substrate or generated product per minute and milligram of protein and mitochondrial content (nmol/minute·mg protein.CS).

Mitochondrial respiration and oxygen consumption rates (OCRs) were measured following the manufacturer’s protocols with two different technologies: Oroboros^TM^ high resolution respirometry [34] and Agilent Seahorse^TM^ XF24 Analyzer (Wilmington, DE, USA) using the Cell Mito Stress Test [35]. Punctual differences in the technical procedures were made, in order to adapt the experimental procedure to each methodology, which are further described in the Appendix A. OCR values were normalized to total cell protein content and mitochondrial mass, measured by CS enzymatic activity (pmol/min.ug protein.CS).

Lipid peroxidation was quantified by the spectrophotometric measurement of malondialdehyde (MDA) and 4-hydroxyalkenal (HAE) as indicators of reactive oxygen species (ROS) damage into cellular lipid compounds, as reported elsewhere [36]. The results were normalized per mitochondrial content (µM MDA and HAE/mg protein.CS).

Mitochondrial morphology was assessed by immunocytochemistry using confocal microscopy, as previously described [8]. Briefly, a minimum of three fibroblasts from each subject were analysed using a semi-automatic custom-made macro [37] for Image J software [38]. The mitochondrial network of each cell was subjected to particle analysis and the following parameters were assessed: aspect ratio (AR) (major axis/minor axis) and form factor (FF), which was calculated as the inverse of the circularity (circ^−1^; 4π·area/perimeter2). AR and FF values correspond to mitochondrial length and branching, respectively. As mitochondria elongate and become more branched, AR and FF values increase as a sign of mitochondrial health [39]. On the other hand, decreased values of AR and FF are indicative of circular, unbranched, isolated, and thus pathologic mitochondria.

### 2.5. Statistical Analysis

Statistical analysis was performed using two softwares: the GraphPad Prism Software (version 8.3.1 for Mac, San Diego, CA, USA, www.graphpad.com) and the Statistical Package for the Social Sciences (SPSS, version 21, IBM SPSS Statistics; SPSS Inc., Chicago, IL, USA). Differences among groups were sought by non-parametric tests after filtering for outlier values in the datasets. Specifically, Kruskal–Wallis and Mann–Whitney U statistical tests for independent samples were used, when required, together with Holm–Sidak comparison. Significance was accepted for asymptotic two-tailed *p*-values below 0.05 (for a confidence interval of α = 95%) and the results were expressed as means ± the standard error of the mean (SEM).

## 3. Results

### 3.1. Targeted Metabolomic Characterization

Measurement of organic acid and amino acid levels in iPD-fibroblasts showed unbalanced metabolic fluxes related to mitochondrial function.

Organic acids related to mitochondrial energetic metabolism were increased in iPD patients, suggesting the deregulation of the intermediary metabolism (Figure 1).

At standard glucose concentration (5 mM), classical biomarkers of mitochondrial diseases such as lactic acid, and the main components of the Krebs’s cycle (citric, malic, succinic, and 2OH-glutaric acid), tended to increase in iPD samples. Similar trends were observed in the metabolites derived from Kreb’s cycle related to amino acid or fatty acid metabolism (as ethylmalonic or glutaric acid) and the biomarkers from free fatty acid β-oxidation (including adipic, suberic, and sebacic dicarboxylic acids). The accumulation of all these metabolites is frequently associated with MRC dysfunction and, specifically the increase in lactic acid levels, with the activation of anaerobic glycolysis in detriment of MRC function.

A high glucose concentration (25 mM) further accentuated such trends, as observed by the significant increase of citric, suberic, and sebacic acids (*p* = 0.01, *p* = 0.03, and *p* = 0.03, respectively). Such an increment suggests a worsened phenotype for iPD-fibroblasts in high-glucose conditions.

The accumulation of all these metabolites feeding mitochondrial metabolism is frequently associated with mitochondrial dysfunction, specifically the increase in lactic acid levels, with the activation of anaerobic glycolysis in detriment of mitochondrial function.

Moreover, as technical controls, nucleotides and organic acids non-related to mitochondrial energy metabolism (including uracil and pyroglutamic acid) were conserved among patients and controls in both glucose conditions.

Similarly, all amino acids related to mitochondrial function were increased in iPD patients, mimicking the same pattern as organic acids (Figure 2).

At standard glucose concentration (5 mM), alanine, glutamate, aspartate, arginine, and ornithine, the classic amino acids related to mitochondrial metabolism, showed trends towards an increase in iPD-fibroblasts, being statistically significant in the case of glutamate and aspartate (*p* = 0.03 and *p* = 0.008, respectively).

Exposure to a high glucose concentration (25 mM) further confirmed such trends, as observed by the significant increase of all mitochondrially-related amino acids (*p*-values between 0.006 and 0.05 cut-offs). Such an increase suggests the worsening of the phenotype in the case of high glucose exposition, as previously observed with organic acid metabolites.

The accumulation of all these metabolites feeding mitochondrial metabolism is frequently associated with mitochondrial dysfunction.

Once again, only mitochondrial-related amino acids were affected, while amino acids not related to mitochondrial metabolism (such as tyrosine or phenylalanine) were measured (as technical controls) and found to be conserved among iPD patients, controls, and glucose concentrations.

Overall, these findings show increased levels of organic acids and amino acids related to mitochondrial function in iPD-fibroblasts, especially when exposed to high glucose concentration, suggesting that impaired mitochondrial function is exacerbated in ‘pre-diabetogenic’ conditions.

### 3.2. Mitochondrial Characterization

Mitochondrial phenotyping at the enzymatic, oxidative, respiratory, and morphologic level confirmed such an hypothesis.

Specifically, MRC enzymatic activities from CII and G3PDH (fed by Kreb’s cycle and-oxidation pathways) tended to decrease in iPD-fibroblasts (Figure 3), probably generating the accumulation of organic acids and amino acids related to mitochondrial metabolism. Interestingly, CIV activity trended to increase in iPD fibroblasts, significantly when exposed to high glucose (*p* = 0.01), probably to overcome CII-G3PDH impairment. As a result of CII-G3PDH MRC reduction in iPD-fibroblasts, oxidative stress levels, also considered a secondary product of MRC function, trended to decrease in iPD-patients. Additionally, in high glucose, iPD cells manifested decreased metabolic plasticity when compared with controls to adapt to different glucose conditions.

Mitochondrial respiration, measured by Oroboros^TM^ (Innsbruck, Austria) and Seahorse^TM^ technologies (Wilmington, DE, USA) (Figure 4 and Figure 5), confirmed the dysfunction of MRC previously observed at the enzymatic and oxidative level.

OCR measures obtained by Oroboros^TM^ technology (Figure 4) showed that basal (or routine) and maximal (ETC) respiration (after uncoupling), reserve capacity, and as ATP-linked respiration trended to decrease in iPD-fibroblasts.

Mitochondrial respiration measured by Seahorse^TM^ technology (Figure 5) confirmed such trends by the decrease of basal respiration, coupling, maximal, spare, and mitochondria working capacities in iPD-fibroblasts.

In all cases, iPD-fibroblasts showed reduced mitochondrial plasticity with respect to controls to adapt to different glucose conditions.

Changes in mitochondrial morphology were expected after subjecting fibroblasts to different glucose concentrations. In line with this, increased glucose concentration significantly decreased the aspect ratio and form factor from control fibroblasts (*p* = 0.001 and *p* = 0.05, respectively), accounting for less elongated and branched mitochondria. On the contrary, iPD-fibroblasts were unable to adapt to high glucose exposition (Figure 6) and showed conserved mitochondrial morphology, regardless of the media, confirming their metabolic and morphologic rigidity.

Overall, fibroblasts from iPD patients showed a disarranged mitochondrial activity and morphology and manifested the inability to adapt to the different glucose conditions, in opposition to fibroblasts of controls, with preserved bioenergetic plasticity.

## 4. Discussion

Idiopathic PD and T2DM are increasingly prevalent diseases that are epidemiologically associated and share some altered molecular deregulated pathways [11,17,40]. Metabolic and mitochondrial dysfunction have been frequently found to be deregulated in both pathologies. T2DM has been suggested to accelerate mitochondrial dysfunction as a result of exhaustion of the bioenergetic metabolism needed to catabolize the excess of glucose. Interestingly, the relevant bibliography on this field stands for the opposite causal relationship; that is, mitochondrial dysfunction would precede (and eventually cause) T2DM [41]. On the other side, T2DM has been suggested to condition iPD progression because of the toxicity that may cause excessive glucose in neuronal functionalism, among others [42]. Consequently, T2DM treatments have been proposed to condition PD progression [43]. However, little is known on the mechanistic and synergic effect that deregulated pathways may have to support epidemiologic and molecular overlapping between iPD and T2DM. The present study aimed to explore whether the hyperglycaemia characteristic of T2DM may worsen the metabolic and mitochondrial phenotype of iPD fibroblasts, thus highlighting how these pathologic features may explain iPD–T2DM comorbidity.

The systemic effects of deregulated metabolism have been previously described by other studies [6,44], and are relevant in the dissection of the molecular pathways that may lead to the development of treatment strategies [22]. However, none of these studies conducted the potential deregulation of these pathways in iPD-T2DM comorbidity. Despite that neural metabolism and mitochondrial function may be different from that of peripheral tissues, the study of metabolomic and mitochondrial function in fibroblasts is currently validated for the study of neurogenerative diseases, including iPD [45]. Fibroblasts have also been used as a widespread cell model for the study of T2DM [46]. The present work was performed in fibroblasts derived from iPD patients that were exposed to different glucose concentrations, in order to mimic diabetogenic conditions and explore the potential worsening of the molecular phenotype.

In standard glucose conditions (5 mM), fibroblasts from iPD patients showed a basal mitochondrial pathological phenotype accompanied by the accumulation of energetic metabolites related to organic and amino acid metabolism. These pre-existent mitochondrial and metabolic dysfunction results in a defective capacity to adapt to stressful situations, such as the increase in glucose concentration, characteristic of T2DM [13,15,18]. In fact, the present findings unveil the poor resilience of iPD fibroblasts when exposed to high glucose conditions, as opposed to controls, which showed wider plasticity in metabolic and mitochondrial performance. When exposed to high glucose concentrations, iPD fibroblasts worsened their basal pathologic metabolic and mitochondrial status.

Metabolic (including mitochondrial) flexibility is defined as the ability to perform efficient switches in metabolism, depending on the environmental demand (feeding/fasting cycles) [47]. Such capacity seems down warded in iPD fibroblasts, in concomitance with increased risk for T2DM development. Remarkably, iPD fibroblasts showed increased lactate production in detriment of decreased mitochondrial oxidative metabolism. Lactate is a classical marker for the diagnosis of primary mitochondrial diseases, but also a sign of upregulated glycolytic metabolism. Interestingly, T2DM patients also show increased levels of plasmatic lactate [48,49], confirming both the rise of glycolytic metabolism and the metabolic co-morbidity between both diseases.

Whether metabolic alterations are causative or a consequence of mitochondrial dysfunction remains elusive. However, the accumulation of those organic acids and amino acids exclusively related to mitochondrial metabolism supports the hypothesis that metabolite accumulation is triggered by deficient mitochondrial function, as described in the integrated mitochondrial stress response [50]. Supposing that mitochondrial dysfunction preludes metabolic disturbances, mitochondrial targets could set the path for novel therapeutics on iPD and T2DM management. For instance, mitochondrial dynamics has been proven to be one of the most important mechanisms for rapid adaptation to the constant changes of environment, including nutrient bioavailability [42]. In neurodegeneration, the capacity of mitochondria to fuse with one another, supporting the better performance of the MRC, is one of the new targets of early treatment [51,52]. The present study shows an absence of mitochondrial adaptation to changes in glucose concentrations, including morphologic adaptations, which may be relevant in terms of efficiency of the MRC, mitochondrial DNA maintenance, and mitophagy. Further studies and novel therapeutic options may address these questions.

Metabolic and mitochondrial alterations have already been reported in several tissues of iPD and T2DM patients [53,54]. However, the present data demonstrate for the first time the aggravation of the metabolic and mitochondrial phenotype from iPD fibroblasts in a high glucose environment, mimicking T2DM. Although a causal relationship between metabolic or mitochondrial alterations and iPD–T2DM comorbidity is not demonstrated herein, its association is indirectly shown by the global pattern of metabolite accumulation in iPD fibroblasts and the lack of mitochondrial plasticity when exposed to high glucose concentrations. Further studies should assess if alternative pathways of glucose metabolism are affected in iPD cells as glucose entrance or insulin resistance.

Metabolic and mitochondrial disturbances in iPD patients may limit energetic fuel to support the rest of the biologic pathways essential for cell survival. Such bioenergetic failure in iPD seems to be aggravated by T2DM comorbidity, worsening cell fate and explaining the comorbidity between both diseases. These outputs set the paths for potential development of novel therapeutic approaches for PD and T2DM targeting mitochondria, which may be tested in the present model of study of fibroblasts. Perhaps an interesting line of investigation is mitochondrial-targeted protectors such as melatonin for early stages in PD [55] or growth hormone (GH) and insulin growth-factor 1 (IGF-1) supplementation in T2DM [56], but further studies are needed in this respect.

Of note, the present study contains some limitations. First, the model of study for iPD–T2DM comorbidity does not recapitulate some of the manifold molecular alterations that occur in iPD and T2DM, such as disrupted lipid metabolism, increased oxidative stress, or increment of inflammatory response. These deregulations are probably present in the target tissue of disease, but not in the current cell model of fibroblasts, probably because of idiosyncratic metabolic and stress response particularities of considered cells [57]. In line with this, fibroblasts are a validated model for the study of both iPD and T2DM, but we cannot discard the possibility of greater metabolic and mitochondrial extent of alterations in other tissues more dependent on aerobic metabolism or directly targeted to the present diseases such as dopaminergic neurons or β-pancreatic cells [49]. The concomitant analysis of fibroblasts from T2DM patients or those presenting both iPD–T2DM diseases would be of high interest in further studies, as the measurement of other metabolic or mitochondrial parameters. Finally, one of the greatest challenges of this model of study, and other human-tissue derived models, is the great variability among the studied populations. This biological variability together with technical fluctuations and reduced sample size probably weakens the seeking of statistical significations in the present study. Perhaps the analysis of larger cohorts of patients and controls would be optimal for drawing conclusions.

In summary, the present findings demonstrate that systemic metabolic and mitochondrial alterations are present in individuals with iPD, accounting for a decreased capacity to successfully adapt to stressful events throughout life, including a sustained increase of glucose concentration in the extracellular environment. Advances on novel therapeutic targets and potential treatments for iPD and T2DM management would be of great value to face these challenging global health problems, to ameliorate the quality of life of patients, and to reduce the associated sociosanitary burden.

## 5. Conclusions

Fibroblasts from iPD patients showed mitochondrial impairment, resulting in the accumulation of organic and amino acids related to mitochondrial metabolism, especially when exposed to high glucose conditions.

Mitochondrial and metabolic defects down warding cell plasticity to adapt to increased glucose bioavailability may explain the comorbidity between iPD and T2DM.

## Figures and Tables

**Figure 1 antioxidants-09-01063-f001:**
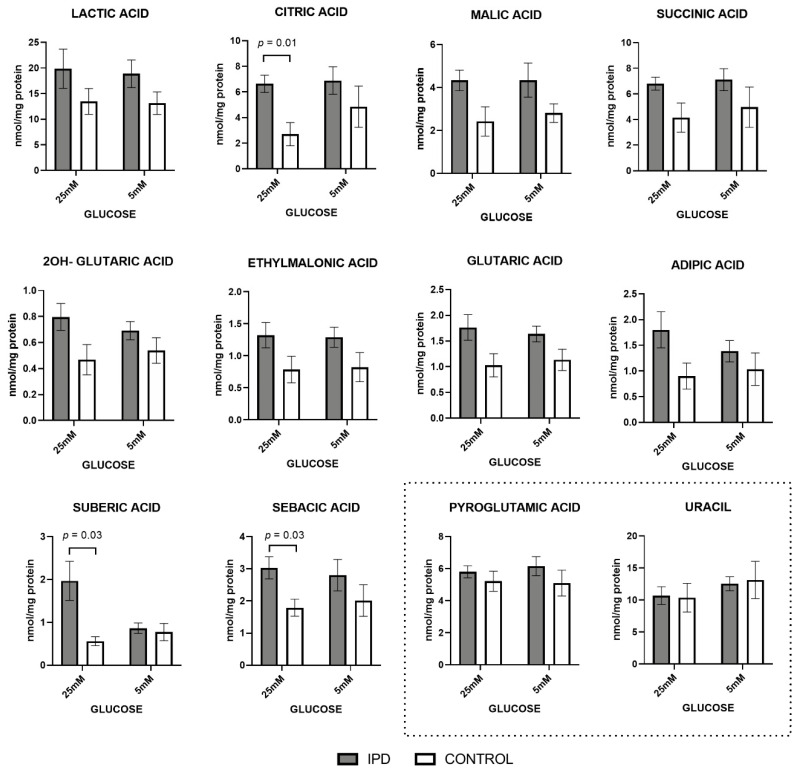
Organic acid levels in fibroblasts of idiopathic Parkinson’s disease (iPD) patients vs. controls exposed to low or high glucose concentration (5 vs. 25 mM). Levels of organic acids related to mitochondrial metabolism were increased in fibroblasts from idiopathic Parkinson’s disease patients (iPD, grey bars) vs. controls (C, white bars), especially in high glucose (25 mM) exposition. As technical controls, the levels of nucleotides and organic acids non-related to mitochondrial energy metabolism (including uracil and pyroglutamic acid) were measured and found to be conserved among patients, controls, and glucose conditions, suggesting that only mitochondrial-related organic acids were affected.

**Figure 2 antioxidants-09-01063-f002:**
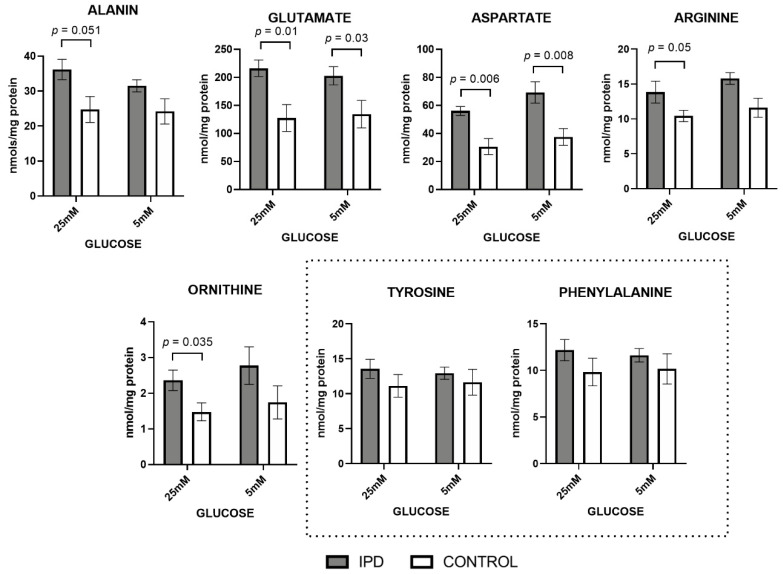
Amino acids levels in fibroblasts of iPD patients vs. controls exposed to low or high glucose concentration (5 vs. 25 mM). Levels of amino acids related to mitochondrial metabolism were increased in fibroblasts from idiopathic Parkinson’s disease patients (iPD, grey bars) vs. controls (C, white bars), especially in high glucose exposition. As technical controls, amino acids not related to mitochondrial metabolism (such as tyrosine or phenylalanine) were measured and found to be conserved among iPD patients, controls, and glucose concentrations, suggesting that only mitochondrial-related amino acids were affected.

**Figure 3 antioxidants-09-01063-f003:**
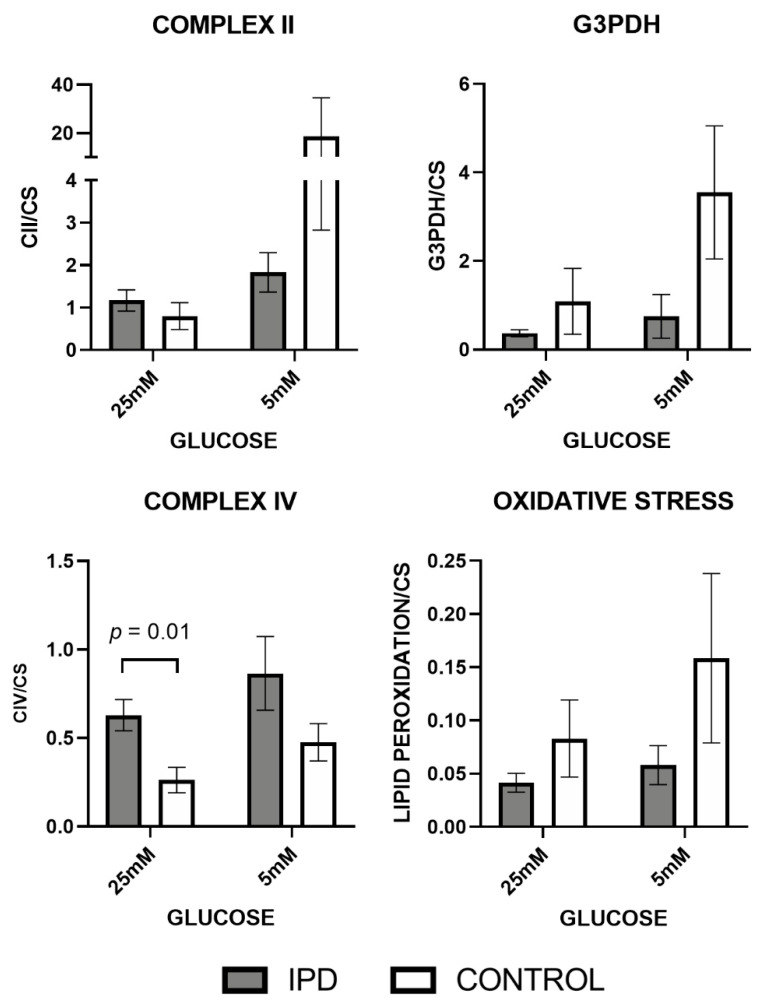
Mitochondrial enzymatic and oxidative stress levels in fibroblasts of iPD patients vs. controls exposed to low or high glucose concentration (5 vs. 25 mM). Mitochondrial respiratory chain (MRC) enzymatic activities were decreased in fibroblasts from idiopathic Parkinson’s disease patients (iPD, gray bars) vs. controls (C, white bars) except for Complex IV (CIV) activity, which trended to increase in iPD fibroblasts, significantly when exposed to high glucose (*p* = 0.01), probably to overcome Complex II (CII)-glycerol-3-phosphate dehydrogenase (G3PDH) impairment. As a result of CII-G3PDH MRC reduction in iPD-fibroblasts, oxidative stress levels, as a secondary product of MRC function, trended to decrease in iPD-patients. In high glucose fibroblasts of iPD patients showed decreased metabolic plasticity than controls to adapt to different glucose conditions. CS, citrate synthase.

**Figure 4 antioxidants-09-01063-f004:**
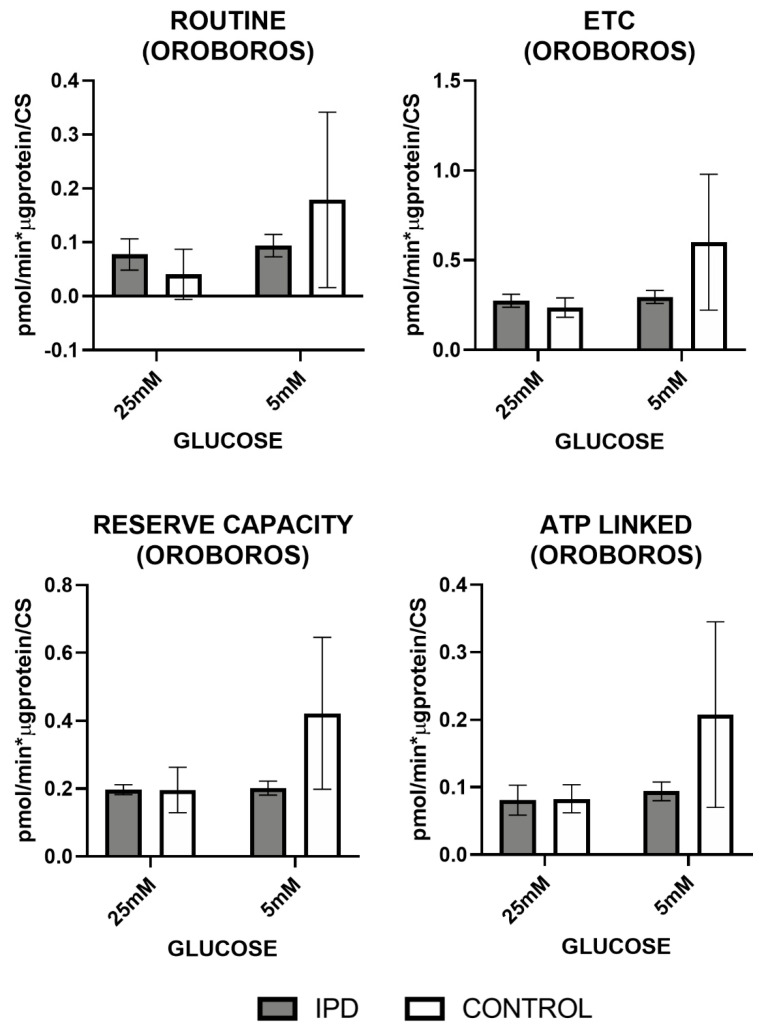
Mitochondrial oxygen consumption by Oroboros technology in fibroblasts of iPD patients vs. controls exposed to low or high glucose concentration (5 vs. 25 mM). Mitochondrial respiration measured through the Mitostress test and Oroboros^TM^ technology confirmed decreased mitochondrial activity in fibroblasts from idiopathic Parkinson’s disease patients (iPD, gray bars) vs. controls (C, white bars), which additionally showed reduced mitochondrial plasticity with respect to controls to adapt to different glucose conditions. ETC: electronic transport chain capacity; ATP: adenosine triphosphate.

**Figure 5 antioxidants-09-01063-f005:**
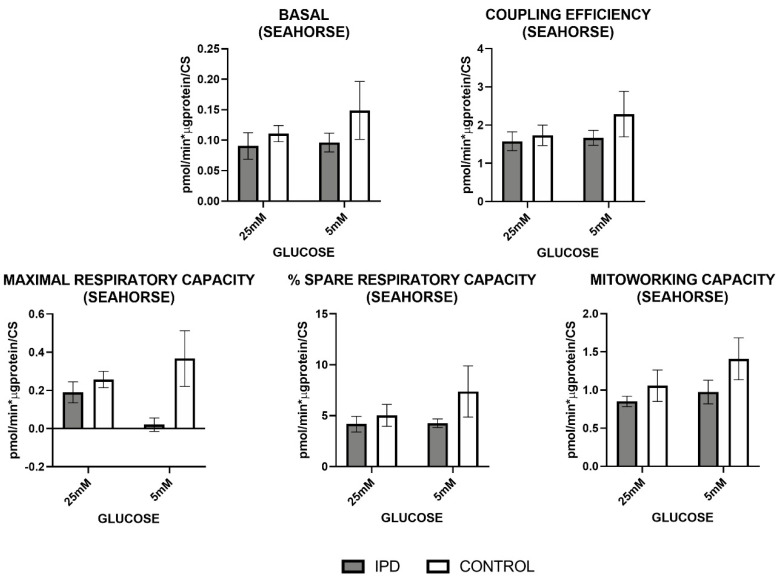
Mitochondrial oxygen consumption by Seahorse in fibroblasts of iPD patients vs. controls exposed to low or high glucose concentration (5 vs. 25 mM). Mitochondrial respiration measured through the Mitostress test and Seahorse^TM^ technology confirmed decreased mitochondrial activity in fibroblasts from idiopathic Parknson’s disease patients (iPD, grey bars) vs. controls (C, white bars), which additionally showed reduced mitochondrial plasticity with respect to controls to adapt to different glucose conditions.

**Figure 6 antioxidants-09-01063-f006:**
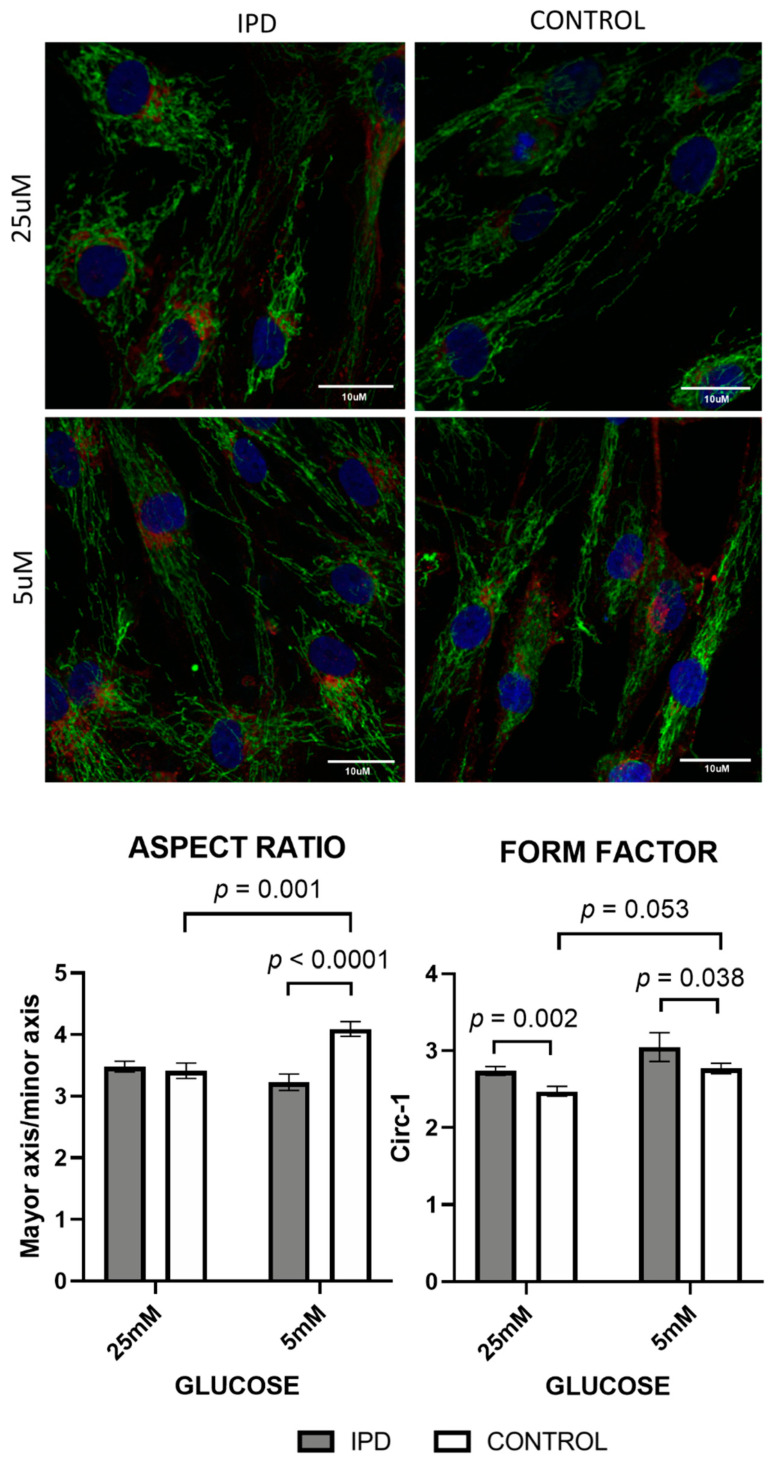
Mitochondrial dynamics (morphology and network) in fibroblasts of iPD patients vs. controls exposed to low or high glucose concentration (5 vs. 25 mM). Upper panel: Representative images of mitochondrial network of fibroblasts of iPD patients vs. controls, in both glucose conditions. Note the preserved mitochondrial mitochondrial porphology and network in iPD cells, compared to controls, able to adapt to different glucose concentrations. Lower panel: Mitochondrial morphology did not change in fibroblasts from idiopathic Parkinson’s disease patients (iPD, grey bars) vs. controls (C, white bars) in accordance with glucose exposition, confirming their metabolic and morphologic rigidity.

**Table 1 antioxidants-09-01063-t001:** Epidemiological characteristics of patients and controls included in the study. SEM, standard error of the mean.

Group	N	Gender		Age		
		Male	Female	Range	Mean	SEM
iPD	7	4	3	46–66	58.57	3.08
Control	7	2	5	41–69	54.86	4.00
Total	14	6	8	41–69	56.71	2.48

No significant differences in age and gender were observed between cases and controls. iPD: idiopathic Parkinson’s disease.

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
