# Peer review of "Disrupted Mitochondrial and Metabolic Plasticity Underlie Comorbidity between Age-Related and Degenerative Disorders as Parkinson Disease and Type 2 Diabetes Mellitus"

_antioxidants, 2020, doi:10.3390/antiox9111063_

Round 1
Reviewer 1 Report
Comments to the author:
The paper by Juarez-Flores DL et al., presents very interesting findings showing the effect of glucose (5 mM and 25 mM) in primary cultured fibroblasts from iPD patients and controls. However, some issues need to be addressed. In general, English grammar and misspelling must be improved.
Major comments
Methods: The Fibroblasts were obtained by a skin punch biopsy and mutation screening was performed. The authors must clarify whether the skin biopsy were made in the same part of the body of all the participants or they just chose randomly the part of the body to make the skin biopsy of all participants.
The authors mentioned that cells were exposed for 10 days to 2 different glucose conditions to assess the metabolic and mitochondrial contribution to disease: a ‘pre-diabetogenic’ high glucose (HG) environment containing 25mM of 115 glucose (equivalent to 450mg/dl) or to a normoglycemic environment containing 5mM of glucose 116 (equivalent to 90mg/dl; low glucose, LG). The authors did not mention whether or not they keep cells with these mediums for 10 days or their changes around every third day, for example. The authors must clarify this part of the cell culture protocol. If the authors did not change the medium continuously, they must justify why they keep for 10 days the cells with the same medium in the discussion/method sections.
Figures 3 and 4: The error bar (5 mM, control) is really big. The authors must discuss this situation, for example, if it is the case that they can increase the n, might it be statistically significant?. Authors are encouraged to discuss the implication of the ¨n¨ in these experiments (n=7).
Figure 6: High glucose (HG) concentration significantly decreased aspect ratio and form factor from control fibroblasts, Authors are encouraged to add picture from this graphic.
There are misspelling and English grammar errors, for example, the Authors used if instead of whether. Line 111 foetal instead of fetal, etc. A grammar correction might improve the paper presentation
.
Minor comments
Figures description.
The authors show very nicely the figures. However, in the legend Figures, the authors should ensure to describe the figure in the legend figure and the results added in results sections.
Reviewer 2 Report
This is a very interesting study in which the authors analyze the mitochondrial function in cultured fibroblasts from Parkinson patients exposed to different glucose concentrations. They conclude that mitochondrial respiratory chain dysfunction in conditions mimicking diabetogenic conditions may explain the comorbidity between PD and type 2 diabetes mellitus.
The study is methodologically correct, results are clear and well presented and the discussion shows the limitations that the study may carry out.
References are correct.
However, this reviewer has some suggestions that, perhaps, the authors should explain or analyze in their work:
1) why they did analyze only factors II and IV of the mitochondrial respiratory chain, and not factors I and III too, as well as the cytochrome c?
2) It would be advisable to analyze the addition of melatonin at different doses to these cultures and check whether melatonin could improve the glucose-dependent mitochondrial affectation. In fact, we treated some Parkinson patients with high doses of melatonin and they experienced very significant positive changes (unpublished results). Melatonin is a mitochondrial protector and able to partially or almost fully replace the lost of function of factors I and III in the mitochondrial respiratory chain.
3) In addition, GH addition to the culture might also be used, because this hormone is also a mitochondrial protector, and there are not problems when treating diabetic patients with this hormone. In fact recent studies showed that GH increases the production of pancreatic ß cells in these patients.
4) a last question, without major repercussion for this study its that the authors said that 90% of PD patients have idiopathic origin. Perhaps this is not a right percentage, because traumatic brain injury leads to a really high number of PD patients, either because of a degeneration of presynaptic afferences to dopaminergic neurons or to a direct destruction of them.
Reviewer 3 Report
This is a very interesting paper in which the authors tried to show that mitochondrial dysfunction is present in iPD and T2DM and that it is the cause of the comorbidity of these diseases. However, in my opinion, with the research model used, it was shown that the mitochondria of fibroblasts collected from patients with iPD present features of dysfunction because their response to high glucose concentration is different than those in the case of fibroblast mitochondria taken from people without iPD (healthy). This is also confirmed by the differences in some parameters tested between fibroblasts from iPD patients and healthy ones cultured in 5mM glucose. The manuscript requires significant changes according to the following remarks:
- lines 92-94 what is the purpose of the study?
- Results/Figure 1 - What do Authors mean in the following sencetnce: "Organic acids related with mitochondrial energetic metabolism were increased in iPD patients, ..." ? In the figure 1, please add statistical analyzes comparing iPD 25mM vs iPD 5mM and control 25 mM vs control5 mM. It is very important to show how the concentration of metabolites is affected by the glucose concentration in normal state (control) and patological state (iPD). Please note the differnecs in case of adipic acid, suberic acid, citric acid and provide relevant comments in the Result section.
- Legend to figure 1 and lines 210-213, uracil is not an organic acid
- Again in Figure 2, please add statistical analyzes comparing iPD 25mM vs iPD 5mM and control 25 mM vs control5 mM. It is very important to show how the concentration of aminoacids is affected by the glucose concentration in normal state (control) and patological state (iPD). Please note the differnecs in case of alanine, asparate,arginine, ornithine and provide relevant comments in the Result section.
- do obtained results allow to provide the answer for the following issue: whether the impaired glucose utilization in fibroblasts derived from patients with iPD is a result of mitochondrial dysfunction
- Again in Figure 3, please add statistical analyzes comparing iPD 25mM vs iPD 5mM and control 25 mM vs control5 mM. It is very important to show how the level of MRC enzymatic activity and oxidative stress is affected by the glucose concentration in normal state (control) and patological state (iPD). Please provide relevant comments in the Result section. How do you explain the increased level of lipid peroxidation marker in 5mM /25mM control vs iPD 5mM /25mM since glucose induces ROS generation, especially high glucose concentration and iPD is associated with increased oxidative stress markers.
- Again in Figure 4 and 5, please add statistical analyzes comparing iPD 25mM vs iPD 5mM and control 25 mM vs control5 mM and provide relevant comments in the Result section.
- The aim presented in lines 304-306 is not clear: This study aimed to explore if metabolic and mitochondrial complications are critical pathologic pathways that may explain this epidemiologic comorbidity. the Authors did not explore pathways, they evaluate the effect of hyperglycemia onmitochondrial function via measurement of selected metabolites,.....and activity of complex II.....
- Discussion: "The present work was performed in fibroblasts derived from iPD patients that were exposed to changing glucose concentrations, to
mimic diabetogenic conditions and explore the potential worsening of the molecular phenotype. "The Authors did not change glucose concentration, they culture fibroblast for 10 days in medium containig 25 mM of glucose that means that the cells were chronically exposed to hyperglycemia. The cells cultured in medium containing 5 mM of glucose for 10 days mimicked the chronically normoglycemic condition. Thus, the term "changing glucose concentrations" is used to indicate that for example within 10 days; for 2 days the cells were cultured in HG, then the medium was changed into new one containing 5 mM for 2 days, and so on... - Discussion- pleas explain what kind of metabolic complications were investigated in the study as suggested in lines: 305-306 "This study aimed to explore if metabolic and mitochondrial complications are critical pathologic pathways that may explain this epidemiologic comorbidity."
- lines:315-317 "In standard glucose conditions, fibroblasts from iPD patients showed a basal mitochondrial pathological phenotype that was accompanied by the accumulation of energetic metabolites related
to organic and amino acid metabolism." Please clearly indicate which results suport above stentence from Discussion. - Discussion - lines 348-349: "A causal relationship between metabolic or mitochondrial alterations and iPD T2DM comorbidity is not herein demonstrated, but its association is indirectly shown. " the associations should be clearly indicated based on the results.
- Please explain in details how T2dm may accelerate mitochondrial dysfunction and iPD progression?
- The discussion is poor, and Authors should use more references that support or not support their findings.
- In my opinion this conclusion: "Mitochondrial and metabolic defects down warding cell plasticity to adapt to changing glucose bioavailability may explain the comorbidity between iPD and T2DM." is not supported by the result presented in the study.
Round 2
Reviewer 3 Report
Dear Authors,
You sufficiently addressed to my comments and the revised version of your paper now is clearer and presents hypothsis supported by the results.
Best regards,